# Key Soft Skills in the Orientation Process and Level of Employability

Ana Rodríguez Martínez [1,*], Verónica Sierra Sánchez [2], Carolina Falcón Linares [1] and Cecilia Latorre Cosculluela [2]

1 Department of Educational Sciences, Faculty of Education, University of Zaragoza, 50009 Zaragoza, Spain; cfalcon@unizar.es
2 Department of Educational Sciences, Faculty of Human Sciences and Education, University of Zaragoza, 22003 Huesca, Spain; vsierra@unizar.es (V.S.S.); clatorre@unizar.es (C.L.C.)
* Correspondence: anaromar@unizar.es; Tel.: +34-653-139-450

**Abstract:** We understand soft skills as an integrated set of knowledge, skills and values that facilitate the effective, affective and efficient development of a skill or activity. The objective of this study was to identify the soft skills that are most often used in the orientation process and those that improve employability, according to Spanish counselors. A qualitative methodology was used. The participants ($n = 57$) were orientation professionals. A total of 273 interviews were conducted over five years. The results and conclusions show that communication and decision-making are the most important competencies in the orientation process and at the level of employability, followed by intrapersonal skills, skills in interpersonal relationships, teamwork, problem solving, adaptation to new situations, creativity and leadership.

**Keywords:** soft skills; decision making; personal and professional development; communication

## 1. Introduction

The management of soft skills determines the orientation process and employability levels of people undergoing orientation. Investing in soft skills implies the prevention of higher levels of unemployment and the promotion of personal and social well-being.

With this research, our goal was to reinforce quality professional guidance that contributes to equal opportunities and favors citizens who need such guidance the most and are in vulnerable circumstances, such as the unemployed.

What are the most important soft skills to promote during the orientation process? What soft skills facilitate employability? These are the questions that we set out to answer in this research, which began in 2016 and was completed in 2020. This research was approved by the Research Ethics Committee of the Autonomous Community of Aragon (Comité de Ética de la Investigación de la Comunidad Autónoma de Aragón–CEICA) in Spain.

Soft skills are a combination of competencies that concern various aspects of the individual and are related to (a) knowledge, aptitudes and technical skills (knowledge); (b) methodological approaches to work processes (knowing how to do); (c) individual and collective patterns and forms of behavior (knowing how to be present); and (d) methods of organization and interaction (knowing how to be) Rodríguez et al. [1]. These competencies are only definable in action, i.e., in work situations, as indicated by the term "professional action competence"; therefore, we represent the culmination of this combination with the addition of the term "action" to labor or professional competence to reflect the need to put theoretical knowledge into practice.

We say that a professional is competent or possesses professional competence when he/she uses the knowledge and skills he/she has learned in his/her training (technical competence) (knowledge). In addition, he/she applies this knowledge to various professional situations and adapts it according to the requirements of his/her work (methodological

competence) (knowing how to do), but this is not enough; to be truly competent, he/she must be able to interact and participate with his/her peers in the team actions necessary for their professional task (participatory competence) (knowing how to be present). Finally, he/she must be able to solve problems autonomously and flexibly and to collaborate in the organization of work (personal competence) (knowing how to be) [2].

Since the 1990s, there has been growing interest from the European Commission and the OECD in determining what values and skills are needed by citizens to improve their levels of employability. Soft skills are those that allow individual to be good citizens as well as good professionals; thus, to improve soft skills in the orientation process, we must implement classical models and complement them with ethical, support and social equality approaches.

The term "competence" is not univocal, and international projects refer to the concept with different terms. The OECD, in the DeSeCo project, used the concept of "competencies"; the Cedefop Project [3] referred to "skills" in the way that Reflex [4] referred to "competences", and in the year 2000, the Tuning project continued this concept and introduced the terms "capacity", "ability", "attribute" and "skill".

As we have seen, there are no unanimous criteria for defining competency, and there are as many definitions as there are international projects and reports [5–9]. It seems that at present, unifying the concept of competence is an impossible mission.

In more than half of the ANECA white papers and degree proposals, general competencies are differentiated from transversal competencies, while ANECA, DeSeCo [5] and Tuning [6] have stated that the only thing that separates general competencies from transversal ones is nomenclature. If general and transversal competences are the same, why, then, are they differentiated in the documents that lay the foundations for constructing degrees and orientations?

At this point, the following questions must be asked: What is the point of differentiating technical competencies from general ones if knowledge and skills are included in general competencies? Soft skills are an infinite resource that must be prioritized to provide equal opportunities and free and accessible learning for all [5,6]; therefore, far from trying to confuse the issue, the purpose of this article is to describe the actual situation, analyze it and attempt to present unified criteria to improve the employability levels of individuals and thus contribute to a more just and equitable society. Hard skills are those that are developed at a specific level in each of the professions and are composed of particular and eminently technical or methodological contents, such as knowing the structure of a didactic unit, knowing how to operate a machine or knowing the side effects of a medication. On the other hand, soft skills are the skills common to all degrees and professions, such as communication, problem solving or decision-making.

To perform this study, we selected the 24 soft skills shared in the DeSeCo (3), Tunning (6) and DeSeCo (5) projects. Following these classifications, we divided the selected competencies into three large blocks: (1) Instrumental, composed of eight soft skills, including (1) Ability to analyze and synthesize, 2. Ability to organize and plan, 3. Oral and written communication in the native language, 4. Knowledge of a foreign language, 5. Computer knowledge related to the field of study, 6. Capacity of information management, 7. Problem solving and 8. Decision making; (2) Personal, composed of eight soft skills, including 9. Teamwork, 10. Interdisciplinary teamwork, 11. Work in an international context, 12. Intrapersonal skills, 13. Interpersonal skills, 14. Recognition of multicultural diversity, 15. Critical reasoning and 16. Ethical commitment; and (3) Systemic, composed of eight soft skills, including 17. Self-learning, 18. Adaptation to new situations, 19. Creativity, 20. Leadership, 21. Knowledge of other cultures and customs, 22. Initiative and entrepreneurial spirit, 23. Motivation for quality and 24. Environmental sensitivity.

With this research, we intend to positively influence advisers and those oriented toward investing in the values of the 2030 Agenda recently proposed by the United Nations. The conditions of the present and the near future imply high levels of ambiguity, change and complexity [10], and therefore, we must take into account the three benefits of

using the concept of soft skills: (1) an awareness of which transversal skills are necessary for the development of most professions, (2) the provision of internationally accepted classifications to be able to use them in the creation of degrees, documents and reports of educational, social and political interest and (3) in this case, specifically in the area of career guidance for employment support, the definition of which skills a counselor should have to, in turn, be able to train and guide their users, because it is difficult for a professional counselor to help manage time if their own time management is deficient. Determining these soft skills offers a roadmap for the selection of guidance professionals.

The objective of this study was to identify the soft skills most used in the Spanish orientation process and compare them with the soft skills that improve employability according to Spanish counselors. The research hypothesis states that in the Spanish orientation process, there are between two and four eminently used soft skills that affect the level of employability of users.

## 2. Materials and Methods

### 2.1. Participants

All counselors who participated in the research provide free and open-access services for anyone who needs them, although the counselors are also mostly people with limited economic resources and are at risk of social exclusion.

The sample comprised 57 professional counselors from Spain.

This study was longitudinal in nature because it took place over five years (2016–2020) through semistructured interviews conducted with the purpose of determining which soft skills have the greatest impact on the orientation process and employability.

A total of 273 interviews were conducted (232 online and 51 face-to-face), and 12 participants were lost over the five-year period.

The sample comprised 33% men and 67% women, and their ages were between 32 and 56 years.

Forty-eight percent had unstable jobs, and 91% had higher education degrees from a university (62%) or vocational training (28%).

Eighty-six percent of the counselors worked in public employment services, and the remaining 34% worked in social projects subsidized by the European Union or social entities.

The counselors had an average of 12 years of experience, with five years as the minimum and 24 years as the maximum.

Twenty-eight percent of the sample lived in the autonomous community of Aragon, 12% lived in Andalusia, 11% lived in Madrid, 10% lived in Castile and León, 8% lived in Castile la Mancha, 7% lived in Basque Country, 5% lived in the Valencian Community, 6% lived in Catalonia, and 9% lived in Galicia. The remaining 4% represented the autonomous communities of Cantabria, Asturias, Navarra and La Rioja.

Seventy-six percent of the people receiving counseling were at risk of social exclusion and used the counseling services to improve their level of employability. There are several main reasons why the Spanish state considers users as being at risk of social exclusion: (1) being long-term unemployed (more than a year in unemployment), having economic difficulties (having minimal income or more expenses than income), sociocultural reasons (recent immigration) and/or the ability to integrate into the business system (due to some type of illness, disability or specific need for extraordinary support).

### 2.2. Information Collection Tool

The tool used to obtain information was a semistructured interview that consisted of four phases: (1) explanation of the research, collection of demographic data and acknowledgement; (2) questions related to the professional career, orientation models and projects of the participants; (3) questions about 24 soft skills (see Table 1 selected from DeSeCo [5] and Tuning [6]; (4) at the end of the interview, conclusions and the scheduling of an appointment for the next interview.

**Table 1.** Soft skills and their level of importance in the orientation process and employability.

| Instrumental Soft Skills | * O | * E | Personal Soft Skills | * O | * E | Systemic Soft Skills | * O | * E |
|---|---|---|---|---|---|---|---|---|
| 1. Capacity for analysis and synthesis | 6.4 | 4.3 | 9. Teamwork | 7.6 | 8.3 | 17. Autonomous learning | 6.3 | 6.2 |
| 2. Capacity for organization and planning | 6.1 | 6.3 | 10. Interdisciplinary teamwork | 4.5 | 7.3 | 18. Adaptation to new situations | 7.2 | 7.1 |
| 3. Oral and written communication in the native language | 9.7 | 9.5 | 11. Work in an international context | 5.1 | 6.7 | 19. Creativity | 7.5 | 7.3 |
| 4. Knowledge of a foreign language | 3.1 | 6.2 | 12. Intrapersonal skills | 8.9 | 9.1 | 20. Leadership | 7.9 | 7.6 |
| 5. Computer knowledge related to the field of study | 6.2 | 7.2 | 13. Interpersonal skills | 8.7 | 9 | 21. Knowledge of other cultures and customs | 4.3 | 4.1 |
| 6. Capacity for information management | 3.6 | 6.8 | 14. Acknowledgment of multicultural diversity | 3.5 | 5.4 | 22. Entrepreneurial effort and spirit | 5.6 | 6.3 |
| 7. Problem solving | 8.1 | 8.2 | 15. Critical thinking | 5.2 | 4.6 | 23. Motivation for quality | 5.4 | 5.3 |
| 8. Decision-making | 9.6 | 9.4 | 16. Ethical commitment | 5.2 | 4.1 | 24. Environmental sensitivity | 4.7 | 3.8 |

Mean importance on a scale of 0 (minimum) to 10 (maximum) for orientation process (* O) and employability (* E); Source: by authors.

The interviews conducted in the research were divided into three large blocks, the first of which was related to self-perception regarding the level of acquisition of soft skills. The second block consisted of deepening in lived experiences in which the user had put soft skills into practice. The third block consisted of an analysis of the context and how to improve the level of employability through these skills. In this phase, the users were encouraged to develop their own academic and professional projects.

In each of the interviews conducted, the interviewee was asked to assess the level of soft skills following two criteria: (1) importance of soft skills in the orientation process and (2) how this competence influences the level of employability of the users served.

*2.3. Initial Configuration of the Tool and the Pilot Test*

This tool was configured according to the usual steps to guarantee its validity [11].

1. An interview was conducted with a group of five experts to clearly define what to measure (competencies) and to determine the most appropriate configuration of the interview.
2. A second review with another group of six external experts was conducted to detect possible errors in the statements or concepts.

Finally, a pilot test was conducted with a small sample of a homogeneous population (18 individuals) to assess the function and understanding of each of the interview questions.

**3. Results**

The evolution of the competencies varied among the different soft skills over the five years of the study. Table 2 shows in detail the average level of importance of these skills in the orientation process and the level of employability.

Seventy-four percent of the counselors who participated in the research considered communication and decision-making to be the key soft skills for improving employability. Based on the perceptions of the counselors and the users, there was an increase in these skills over the five-year study period. The selection of communication and decision-making was justified based on the fact that they are considered key to the development of the specific aforementioned competencies. Communication is basic and transverse to other competencies; it is explored in such statements as "*Participant 31: If you do not know how to*

*communicate with yourself and with others, the rest of the soft skills will be hindered"; "Participant 18: Communication helps with problem-solving, teamwork, work in international contexts and many other soft skills"; "Participant 7: Decision-making involves action and being proactive, and users who are able to make appropriate decisions find work and/or maintain the work they already have much more easily"; and "Participant 24: I am increasingly surprised by the importance of decision-making in the orientation process and employability. Sometimes users do not realize that not making a decision is in fact making one from a passive standpoint. Action is decisive, and decision-making helps to draw up organized and effective plans."*

**Table 2.** Sample.

| Characteristic | Group | Percentage (%) |
|---|---|---|
| Age | 30–35 | 12 |
| | 35–40 | 47 |
| | 40–45 | 28 |
| | 50–56 | 13 |
| Gender | Women | 67 |
| | Men | 33 |
| Employment situation | Public services | 86 |
| | Private services | 34 |
| Type of studies | Graduates | 76 |
| | Masters specializing in the field of orientation | 34 |
| Autonomous community | Aragon | 28 |
| | Andalusia | 12 |
| | Madrid | 11 |
| | Castile and León | 10 |
| | Castile la Mancha | 8 |
| | Basque Country | 7 |
| | Valencian Community | 5 |
| | Catalonia | 6 |
| | Galicia | 9 |
| | Other | 4 |

Table 2 shows the classification of soft skills and their importance for improving employability based on the average findings from the five years of research.

The five most important soft skills for the orientation process are communication (9.7), decision-making (9.6), intrapersonal skills (8.9), interpersonal skills (8.7) and problem solving. The five most important soft skills for employability are communication (9.5), decision-making (9.4), intrapersonal skills (9.1), and interpersonal (9) skills and teamwork (8.3).

We found that teamwork takes on greater importance for employability than in the orientation process, although there are four competencies that are key for both the orientation process and employability.

The least valued competencies in the orientation process are knowledge of a foreign language (3.1), acknowledgement of multicultural diversity (3.5) and the ability to manage information (3.6). The soft skills that are the least valued for the level of employability are environmental sensitivity (3.8), knowledge of other cultures and customs (4.1) and ethical commitment (4.1) (see Figure 1).

It is worrisome that environmental sensitivity, knowledge of other cultures and ethical commitment had such low scores, and this finding invites reflection on possible reasons; a new line of research may need to be opened for a deeper exploration of this topic.

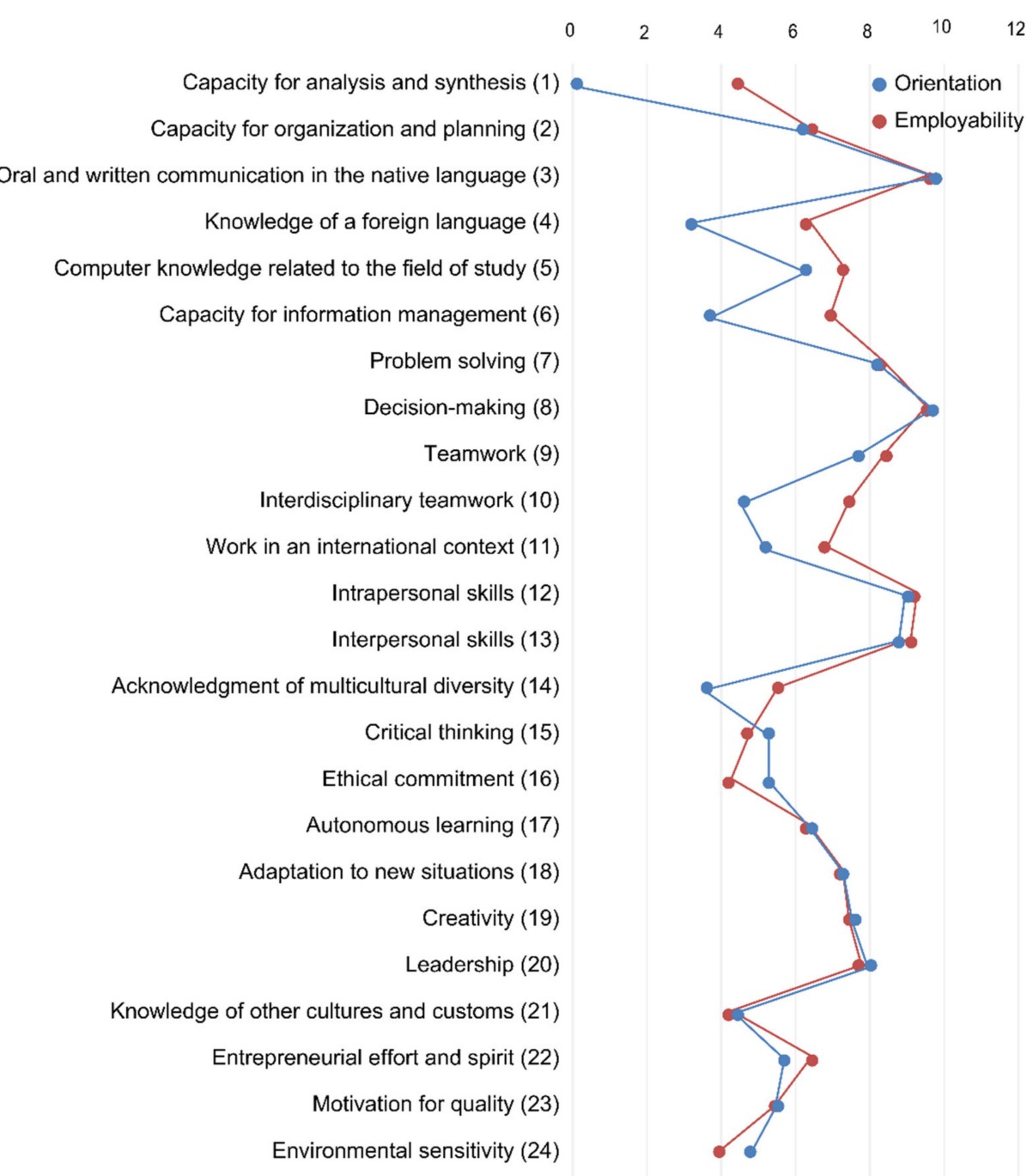

**Figure 1.** Importance of soft skills for orientation and employability.

While men sustain success with factors related to organization and planning, proactivity and the capacity for analysis and problem solving, women focus on maintaining success through aspects related to communication, quality, useful decision-making and capacity for adaptation. Both men and women agree on the importance of self-knowledge and balance in decision-making [12]. Below, we show some of the reflections from the interviews that reflect this distinction between genders: "*Participant 24: Based on the 12 years of experience I have as a counselor I have realized that I and the men I guide focus on eminently practical issues such as organization, management and planning. Participant 56: The men I guide are primarily concerned with solving a problem and working with the guidelines for action. Participant 31: Men are eminently planners and have a capacity for analysis that is very focused on improvement or economic results. Participant 18: The men I guide have a different pattern of behavior than women;*

*they focus more intensely on practical aspects and take into account the context, conciliation and social relationships. Participant 23: While men tend to have economic remuneration as a priority, women tend to focus more on the workday and the type of company. Participant 37: The differences between men and women are fundamentally related to vital objectives; men have a greater vision in the short term, while women have it in the long term.*" Based on this finding, we can state that women have a preference for intuitive decision-making, while men have a preference for analytical decision-making. These competencies are transferable and are learned, and they are equally prone to being unlearned if they are not constantly worked on from a standpoint of flexibility and effort [12].

Some of the improvements proposed by the counselors were developing a greater understanding of the decision-making process and working on communication by elaborating on discourses that could help in a job interview or in the advancement of a product or project. The counselors expressed the importance of continuous training and experience sharing and in having time to provide quality guidance without pressure.

Demotivation, the daily emotional burden and a lack of preparation were handicaps reported by the counselors.

The counselors observed that communication with oneself and with others largely determines an individual's motivation, self-esteem and energy when looking for a job.

Decision-making and communication are the skills that most increase the probability of finding or keeping a job. The counselors stated that these soft skills are key in the orientation process and in the personal and professional trajectory of the user.

Decision-making is inherent to human life and experience. We make an average of 20,000 decisions a day. More than half of the decisions we make daily are automated. An individual's skill in selecting the best option among several determines his/her intelligence [13].

Choosing among several options can be a very simple task at times, but sometimes it is extremely complex and raises concerns. Decision-making involves numerous cognitive processes, including processing of the stimuli present in the task, recall of previous experiences and estimation of the possible consequences of different options. These processes involve working memory and the so-called executive function [14].

However, current research increasingly emphasizes that decision-making is not a mere rational process of accounting for or comparing the gains and losses resulting from a given choice. Rather, it seems that emotional factors, derived from one's own experience or the experience of others in similar situations and factors associated with the consequences or the context in which the decision is made play a determining role [15].

Decision-making is a process that people undertake when they must choose among different options. The decision-making process involves responding to a stimulus (internal or external). This process has different levels of complexity. The level of complexity is determined by the individual, who decides and/or chooses among the different options that are proposed.

Janis and Mann [16] asserted that decision-making is a process, adding that this process involves a stress-generating decision conflict. The stress of decision-making is fueled, according to the authors, by two sources: (I) concerns about objective losses and (II) concerns about subjective losses. The individual experiences a strong desire to end the problem as soon as possible, and the desire to make a decision (prematurely bringing closure to the problem) clashes with an equally intense desire to avoid or at least postpone any decision (causing the problem to linger).

Numerous studies have shown that the role of emotions in decision-making is critical and necessary. Researchers such as Damasio et al. [17] of the University of Pittsburg showed that 600 milliseconds before we think, we feel. Emotion suggests the course of action, which later justifies cognition; this premise is similar to the statement that emotion decides and reason justifies [15,18].

A stimulus causes an unconscious and automatic drive, and this biochemical drive is generated in regions that store information about similar situations experienced in the past.

We repeat learned patterns. The probability of making the same decision when faced with a stimulus that is the same as or similar to a previously received one is very high, although flexibility in decision-making, the power to plan our future and valuing losses or gains are determined by our cognition and how it choreographs emotional stimuli. Our emotions set the direction, and cognition chooses the path.

According to Rodríguez et al. [19], decision-making is the eighth instrumental competency. This block of competencies encompasses skills related to communication, technology, organization and the ability to solve problems and make decisions. These competencies are characterized by a high cognitive content, although their relationship with the other competencies is holistic because all competencies behave and act within a systemic network.

*Good Practices among Counselors for Developing Decision-Making and Communication in Counseling Processes*

Next, we present some of the best practices used by the counselors to promote the soft skills that were most often addressed during the guidance processes and were most valued for improving employability.

For daily decisions, counselors do not advise investing time and energy unless the objective is to change a habit (Participant 15. On numerous occasions, users brood over situations that are not significant and that will not change their lives; Participant 17: Guidance in focusing attention on relevant decisions is key; Participant 43: I do not recommend losing time in making trivial or insignificant decisions; and Participant 35: There are occasions in which chance plays an important role). Instead, they invest time and energy in transcendental decisions and those that represent an important change. For these situations, they propose the following steps:

(1) Define the problem specifically and in detail. (*Participant 19: I consider it essential to make the problem clear from the beginning; Participant 51: When making decisions, what works best for me is to define what we are talking about and write down the needs; and Participant 13: Specifying what the problem is, is the base of a good posterior structure*)

(2) State all possible alternatives. (*Participant 16. Stating where we started and what the options are helps a lot in the decision-making process; and Participant 3: being clear about the alternatives is fundamental*)

(3) Evaluate pros and cons and reduce options, establish hierarchies and note consequences. (*Participant 14: Once you have defined, described and analyzed the situation, what works best for me is assessing pros and cons and reducing options; and Participant 27: When making decisions we have to value pros and cons, even sorting by preference*)

(4) Choose the best of all options. (*Participant 8: Once you have evaluated all the possible options, you must choose the one that provides the most benefits; and Participant 42: Choosing the best option is necessary*)

(5) Select an action and execute it. (*Participant 14: It is not important to know what to do but rather to be able to do it; and Participant 19: Taking action is the execution phase*)

(6) Evaluate the results of the decision that was made and use the experience to inform possible future decisions. (*Participant 28: I consider it essential to perform a meta-evaluation or an evaluation of the process itself; Participant 47: Evaluating the decision-making process will help you to not make the same mistakes*)

Based on the interviews conducted, the counselors stated that there are two fundamental aspects of decision-making to consider during the orientation process: (1) Time. We do not have enough time to assess the pros and cons of each option during the orientation process. (2) Invested energy. Instinctively, we select which decisions require the most energy and attention from us and which decisions can be made directly with the users or students. Some decisions, because of their emotional load, require highly variable investments of energy.

Based on the results obtained and collected in the interviews, the vast majority of counselors defended the idea of interspersing emotion and reason (*Participants 27, 13, 46 and 31*) or intuition and analysis (*Participants 17, 28, 34 and 52*). Counselors work on

two models of decision-making: (1) intuitive decision-making and (2) analytical decision making [20]. This dichotomous view of decision-making involves emotional (intuitive) and rational (analytical) processes [21,22]. (*Participant 22: Instinctive impressions and cognition must be present in the orientation process*).

Information and its planning and analysis constitute the raw material for analytical decision-making; this type of decision-making is sometimes slow, serial, generalized and unreliable. It requires attention and mental effort. This type of decision-making is predominant among males [23]. (*Participant 26 (male): As a counselor, I have the ability to let myself get carried away by the instinctive part and help in what the user needs*).

The findings of Handa [24] and Newman [25] show that intuitive decision-making corresponds to a relational analysis that is crystallized in habit and in the ability to respond quickly through a pattern of experiential recognition [20,24–27]; it results in an effective process for making decisions in complex and uncertain contexts [28]. (*Participant 39 (male): the experience marks my way of counseling, and I try to remember that the users rely on their experiences to achieve their goals*).

In the educational field and during orientation, it is important to focus on methodologies and evaluation processes that allow the integration of intuitive and analytical decision-making systems [26,29]. (*Participant 19: I have not yet carried out any orientation process in which I have only had to respond to one type of decision; for me, both reason and intuition play a priority role*).

Following the empirical studies of Cañabate et al. [30] and Bisquerra and Pérez [31], intuitive decisions yield a higher level of optimization of results than rational decisions because the initial decision is based on emotion and previous experience, while reason provides the necessary time to justify and corroborate the decision that was made instinctively. Intuitive people include affective reactions in their decision-making choices, while deliberate or analytical people refrain from doing so [32,33]. (*Participant 23: When you have experience in counseling, you realize what type of decisions are more likely in a user, and it is easy for you to guide them in adapting to their type of decision-making, although the most effective way is to manage both*).

Epstein [21] states that there are two parallel systems of information processing, one of an analytical type and the other of an intuitive type. Intuition acts automatically and experientially (System 1) and is intimately associated with emotional aspects; it consolidates a spontaneous, rapid and parallel thought process, the input of which comes from knowledge stored in long-term memory (LTM). In contrast, the cognitive system (System 2) corresponds to a slow, serial process of analysis that demands a higher level of mental effort; it is deliberately monitored and controlled, relatively flexible and subordinate to logical-mathematical rules [32], cited by [34]. (*Participant 5: The expectation that the user has regarding employability conditions him; a favorable expectation will lead to positive conditioning and an unpleasant expectation will lead to negative conditioning*).

For counselors, communicating is more than the transmission of ideas from one person to another. It is the means through which we come to know who we are and who we could become [35]. (*Participant 12: Communication is the most powerful tool of counseling*).

After analyzing the different communication styles, techniques and processes used by the 57 counselors who participated in the research, we summarized their communicative procedures in three guidelines that they used to guide people in their personal and professional development processes.

(1)   Active listening.

   a.   They devote time to and validate discourse. They accompany and validate the discourse. (*Participants 7, 18, 54, 25, 36 and 49: We have to validate the emotion of the user. The orientation does not consist of telling others what they have to do but in guiding them to make their own decisions*).

   b.   They delve into the needs of the person they are serving. (*Participants 21, 13, 18 and 45: Maieutic, powerful questions, active listening and the intention to specify what the user wants help them to achieve objectives*).

(2)    They specify the objective in a specific way through positive questions and without making judgments.

(3)    They specify the what, the how, the where and the why of the objectives proposed in the orientation session. (*Participants 54, 29 and 35: Answering the what, how, where and for what helps to establish the academic and professional project of the user*).

According to the DeSeCo, Tuning, Cheers, Reflex and HEGESCO projects, communication is one of the most in-demand instrumental skills in the labor market. Coaching involves communicating emotionally, and coaches are the first to be trained in this competence.

Good communication is key in orientation processes. Although counselors listen more than they speak, when they do speak, they must ask questions and construct their speech very well to provide quality guidance.

A good counselor is able to choreograph the what, the how and the when with ease to ensure that the best possible decision is made.

These results contribute to the objective of this study, which is to know which soft skills are most used in the orientation process and which improve employability according to Spanish counselors. As we proposed in Section 1, the research hypothesis was verified, and we can affirm that, after the analysis of the data, in the orientation process as it is carried out in Spain, there are two eminently used soft skills that affect the level of employability of users: communication and decision-making.

## 4. Discussion and Conclusions

The orientation process is decisive for increasing an individual's level of employability.

It is essential that counselors show fluency in their acquisition of communication, decision-making and intrapersonal and interpersonal soft skills because they act as models during the orientation process, and these are the competencies that counselors should use most often in their own orientation process.

Counselors should not only be competent in these areas but should also have the resources and tools to guide others in acquiring these competencies. The employability of the people who receive counseling will be conditioned by these competencies.

Orientation is not about providing information about resources; it is about actively listening, empathizing and providing excellent communication to help. During the orientation process, it is necessary to show confidence and provide practical resources to facilitate the decision-making process and emotional management.

Communicating with emotion and being attuned to those being served will condition the outcome of the orientation process because, according to Enggruber [36] and Kahneman and Frederick [34], the verbal and nonverbal communication of a counselor can help or hinder the orientation process.

Choosing the best or worst of the options that are presented during the orientation process and in life is determined by the synergy between the (1) intuitive and (2) analytical models of decision-making. Having a healthy network of connections between the two types of decision-making guarantees the ability to make fast and flexible decisions. The speed of decision-making is determined by the type of decision: (1) Intuitive decisions have their origin in the reptilian and limbic brain and are linked to the amygdala and the secretion of adrenaline and cortisol. The flexibility of decision-making is determined by the type of decision. (2) Analytical decisions originate in the cortical brain and are linked to cognitive areas and dopamine secretion.

Beyond showing the key roles of communication, decision-making and emotional skills in the orientation process, the aim of this study was to show how soft skills can be used as an instrument for social equality.

Strengthening these skills through counseling for people who are at risk of social exclusion can improve employability and prevent the development of major problems.

Based on the results of this research, we propose that in practice, counselors should be trained in the development of communicative competence and decision-making because these are key soft skills for the development of employability. According to the results

of this study, it would be convenient to take the perspective of professional counselors and insist on the development of the soft skills that they consider key in the orientation process. The application of this study in practice can be carried out through maieutic techniques, using counseling and orientation strategies such as coaching and neurolinguistic programming.

The limitations of this study include the fact that it was 5-year longitudinal study, which required an immense effort to maintain the sample and to perform active monitoring. The loss of subjects was due to transfers in the workplace and dismissals or terminations of contracts throughout these 5 years. The COVID-19 situation decreased the opportunity to conduct more face-to-face interviews. In addition, the dispersion of the sample is a limitation because each autonomous community had different protocols, and adapting to all of them led to additional wear. Finally, the rapid changes that emerged from the evolution of the system itself could have led to obtain different results at different times. Soft skills evolve, as does the labor market, and we must be quick and aware of the importance of quickly adapting and applying improvements.

As a research prospective, we intend to expand the sample, make comparisons at the international level and design and carry out specific training for professional counselors based on the development of soft skills such as communication and decision-making.

**Author Contributions:** Conceptualization, V.S.S. and A.R.M.; formal analysis, V.S.S., A.R.M.; methodology, A.R.M. and V.S.S.; writing—review and editing, V.S.S., C.F.L., C.L.C. and C.F.L.; supervision, V.S.S. All authors have read and agreed to the published version of the manuscript.

**Funding:** This research received no external funding.

**Institutional Review Board Statement:** Approved by CEICA (CODE: 19/2020 and DATE OF APPROVAL: 07/10/2020).

**Informed Consent Statement:** Informed consent was obtained from all subjects involved in the study.

**Data Availability Statement:** The data presented in this study are available on request from the corresponding author. The data are not publicly available due to privacy and ethical restrictions.

**Conflicts of Interest:** This research is part of an extended collaboration. Supervision was provided by Universidad de Zaragoza, Educaviva grupo de investigación and Aragón Research Ethics Committee (Comité de Ética de la Investigación en Aragón, CEICA), which is officially accredited in Spain. The authors declare no conflict of interest.

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
