# Peer review of "Key Soft Skills in the Orientation Process and Level of Employability"

_sustainability, doi:10.3390/su13063554_

Round 1

Reviewer 1 Report

  1. This is a meaningful and interesting study.
  2. The layout must follow the journal’s rule.
  3. A paragraph consisting of one sentence should be avoided. Please check:

          Lines: 57~59, 60~62,79~80, 87~83, and so on.

  1. Line 87 missing a dot.
  2. In conclusion, the section should be more statement related to the findings.
  3. In the discussion, could add some contents related to how the results of this research are applied in practice.

Author Response

Before explaining all the revisions made, we would like to thank the reviewer for the effort and work put into the article. The revisions made have undoubtedly improved the quality of the final article.

We attach the article including the revisions made in red.

Thanks again and have a great day.

This is a meaningful and interesting study.

Thank you very much for the comments.

The layout must follow the journal’s rule.

In response to the suggestions of the reviewer, we have adapted the design to the rules of the journal.

A paragraph consisting of one sentence should be avoided. Please check: Lines: 57~59, 60~62,79~80, 87~83, and so on.

As has been suggested, the content has been combined to avoid the repetition of short sentences throughout the manuscript.

Line 87 missing a dot.

In conclusion, the section should be more statement related to the findings.

Following the advice of the editor, a global review of the article has been conducted to clarify the findings.

In the discussion, could add some contents related to how the results of this research are applied in practice.

In the discussion, content related to how the results of this research can be applied in practice has been added.

Reviewer 2 Report

The manuscript “Key Soft Skills in Orientation Processes and Levels of Employability” presents a very relevant research topic, through a longitudinal research design. The participants are a critical population for achieving a better understanding of the research problem. I provide, in the text below, some comments and suggestions that may help strengthen the manuscript.

In the fourth paragraph of your paper “Soft skills are represented as the combination of competencies that concern various aspects of the individual related to…” you pointed out a set of specifications that makes it difficult to differentiate soft skills from other types of skills. Can you explain better what distinguishes soft skills from hard skills for instance?

Besides the confusion presented in the official reports and literature regarding the terminology what is the benefit of using the concept of soft skills in general, and specifically in the area of professional consulting for employment support?

It is not clear why the professional “counselors are mostly people with limited economic resources and at risk of social exclusion”. Is this a specific circumstance of your sample?

To help the reader to fully understand your sample characteristics you may add a table that clarifies the characteristics of the sample that provide complete data and the part that drops out. It would be informative to add also the observed frequency.

In your interview protocol you had already chosen 24 soft skills it will be important to explore this option a bit further in the introduction.

Were the participants asked at any moment of the interview about skills or other aspects that they thought critical for employment? 

When the table is designed by the authors it is not probably necessary to add the following information in the legend: “Source: by authors”.

In the results when you write “There was an increase in these skills over the five-years study period” you meant in the perception of the importance of these skills?

It will help to read the quotations if you distinguish between the text and the participant to whom the statement belongs. It may also help to separate the quotation by skill, like putting all the quotations of communication, and then decision making, and so on.

Table 1 is displaying results, so it maybe be better to be put in the results section.

It is not clear how the mean values presented in the table were collected. Were the participants asked to rate the skills?

It would be useful to add the results obtained when dividing the consolers by gender and also the rationale for trying to acknowledge these differences.

The result section would be improved by separating the literature review and some explanation, limitations and reflections that can be placed in the introduction or the discussion and conclusion sections.

In the results (3.1) you state that based on what the consulters shared during the different interviews it is proposed a process for promoting soft skills. Is it right? When you write “For these situations, they propose the following steps”, may you give the reader support from interviewer’s quotations?

The whole sub-section “Good Practices among Counselors for Developing Decision-Making and Communication in Counseling Processes” need to be supported in evidence from the data collection process otherwise it seems that is based in the literature review that you are citing thorough the text.

It is important to discuss, in the last section of your paper, the implication of your results and also the limitations and the future venues of research you are opening with this results.

Author Response

Before explaining all the revisions made, we would like to thank the reviewer for the effort and work put into the article. The revisions made have undoubtedly improved the quality of the final article.

We attach the article including the revisions made in red.

Thanks again and have a great day.

The manuscript “Key Soft Skills in Orientation Processes and Levels of Employability” presents a very relevant research topic, through a longitudinal research design. The participants are a critical population for achieving a better understanding of the research problem. I provide, in the text below, some comments and suggestions that may help strengthen the manuscript.

Thank you very much for all the comments.

In the fourth paragraph of your paper “Soft skills are represented as the combination of competencies that concern various aspects of the individual related to…” you pointed out a set of specifications that makes it difficult to differentiate soft skills from other types of skills. Can you explain better what distinguishes soft skills from hard skills for instance?

To clarify the text and terminology, the difference between hard and soft skills has been explained in greater detail in the introductory section.

Besides the confusion presented in the official reports and literature regarding the terminology what is the benefit of using the concept of soft skills in general, and specifically in the area of professional consulting for employment support?

In response to the improvement proposal, an explanation of the benefit of using the concept of soft skills in general and specifically in professional consulting for employment has been introduced in the text (introduction).

It is not clear why the professional “counselors are mostly people with limited economic resources and at risk of social exclusion”. Is this a specific circumstance of your sample?

Professional counselors are not at risk of social exclusion, but they work with people who show a risk of social exclusion due to being long-term unemployed, having economic and sociocultural difficulties and/or integrating into the business system.

Following the advice of the editor, we have provided a broader explanation at the end of section 2.1.

To help the reader to fully understand your sample characteristics you may add a table that clarifies the characteristics of the sample that provide complete data and the part that drops out. It would be informative to add also the observed frequency.

As suggested by the reviewer to clarify the characteristics of the sample, we have added a table at the end of section 2.1 that reflects the most significant and remarkable data.

In your interview protocol you had already chosen 24 soft skills it will be important to explore this option a bit further in the introduction.

Following the advice of the reviewer, an explanatory paragraph has been added to the introduction to clarify the selection of soft skills and to further explain the selection and classification.

Were the participants asked at any moment of the interview about skills or other aspects that they thought critical for employment?

To answer this question, an explanatory paragraph on the structure of the interviews has been added in section 2.2.

When the table is designed by the authors it is not probably necessary to add the following information in the legend: “Source: by authors”.

Thank you for the suggestion. We have eliminated this legend.

In the results when you write “There was an increase in these skills over the five-years study period” you meant in the perception of the importance of these skills?

This section refers to the perceptions of the counselors and the users. We have clarified this part.

It will help to read the quotations if you distinguish between the text and the participant to whom the statement belongs. It may also help to separate the quotation by skill, like putting all the quotations of communication, and then decision making, and so on.

As the reviewer suggested, citations have been included throughout the text to endorse the explanation; in addition, we have separated the citations by ability, and we have ordered all the communication citations as well as the decision-making citations.

Table 1 is displaying results, so it maybe be better to be put in the results section.

As suggested by the reviewer, the previous Table 1 (currently Table 2) has been included in the results section.

It is not clear how the mean values presented in the table were collected. Were the participants asked to rate the skills?

In each of the interviews conducted, the interviewee was asked to assess the level of soft skills following two criteria: 1) the importance of soft skills in the orientation process and 2) how this competence influences the level of employability of users.

We have provided this clarification in section 2.2.

It would be useful to add the results obtained when dividing the consolers by gender and also the rationale for trying to acknowledge these differences.

Following the advice, we have included the results (section 3) of the interviews conducted to endorse the coherence between what the counselors said and what we found in the literature.

The result section would be improved by separating the literature review and some explanation, limitations and reflections that can be placed in the introduction or the discussion and conclusion sections.

As suggested, in the results section, the literature review has been separated, and some explanations, limitations and reflections have been placed in the discussion and conclusion section.

In the results (3.1) you state that based on what the consulters shared during the different interviews it is proposed a process for promoting soft skills. Is it right? When you write “For these situations, they propose the following steps”, may you give the reader support from interviewer’s quotations?

Following the advice of the reviewer, comments from the counselors who supported the statements and good practices have been added.

The whole sub-section “Good Practices among Counselors for Developing Decision-Making and Communication in Counseling Processes” need to be supported in evidence from the data collection process otherwise it seems that is based in the literature review that you are citing thorough the text.

As suggested by the reviewer, we have incorporated evidence that supports the good practices section.

It is important to discuss, in the last section of your paper, the implication of your results and also the limitations and the future venues of research you are opening with this results.

The implications of the results, the limitations of this study and future research prospects have been included in the last section.

Reviewer 3 Report

Not clearly identified:
research objectives
research hypotheses and how the interview guide was elaborated in relation to the established hypotheses
on a very dynamic labor market, the 5-year research could be marred by rapid changes in the labor market. The required skills can change quickly.

Author Response

Before explaining all the revisions made, we would like to thank the reviewer for the effort and work put into the article. The revisions made have undoubtedly improved the quality of the final article.

We attach the article including the revisions made in red.

Thanks again and have a great day.

Not clearly identified:

research objectives

research hypotheses and how the interview guide was elaborated in relation to the established hypotheses

on a very dynamic labor market,

Following the advice of the reviewer, the objectives and research hypotheses have been included in the final section of the introduction and are addressed in the results section.

the 5-year research could be marred by rapid changes in the labor market.

The required skills can change quickly.

We have noted as limitations the need to react to the dynamism of the labor market as well as the constant changes and their impact on the evolution of soft skills.

Round 2

Reviewer 2 Report

The article explores a very relevant scientific topic and the authors had made some important review that added clarity and a broader context to the manuscript.

I would suggest to add only a few clarifications to the final manuscript, as you can see below.

  1. The first bibliographic reference [1] has the name of the author before (I think it may be error);
  2. In the first sentence of the participants section you state: “All counselors who participated in the research provide free and open-access services for anyone who needs them, although the counselors are also mostly people with limited economic resources and are at risk of social exclusion”.

As you already explained (in the same section) that the people who are in risk are the one who receive the counselors advise you may cut or reformulate the last part of this sentence.

  1. In the result section you write that "Table 2 shows the classification of soft skills and their importance for improving employability based on the average findings from the five years of research". However, table 2 shows sample characteristics (as suggestion in the legend of the Table you could clarify that the information there is about sample description).
  2. Regarding the report of the descriptive statistics why not say the number of cases (observed frequencies) and not just the percentages?
  3. It would be preferable that in page 7 (line 231 to 235) you use the expression like for instance "it seems" or "these perceptions may indicate". Given the sample size and the design of the study, although you have very important evidences, it will benefit the section if the expression to used express more careful.

Author Response

Dear reviewer, I think there has been a confusion with the revised article, since it does not correspond to the submitted work. We attach the improvements made and we are awaiting your response.
Sincerely

Ana

Before explaining all the revisions made, we would like to thank the reviewer for the effort and work put into the article. The revisions made have undoubtedly improved the quality of the final article.

We attach the article including the revisions made in red.

Thanks again and have a great day.

The manuscript “Key Soft Skills in Orientation Processes and Levels of Employability” presents a very relevant research topic, through a longitudinal research design. The participants are a critical population for achieving a better understanding of the research problem. I provide, in the text below, some comments and suggestions that may help strengthen the manuscript.

Thank you very much for all the comments.

In the fourth paragraph of your paper “Soft skills are represented as the combination of competencies that concern various aspects of the individual related to…” you pointed out a set of specifications that makes it difficult to differentiate soft skills from other types of skills. Can you explain better what distinguishes soft skills from hard skills for instance?

To clarify the text and terminology, the difference between hard and soft skills has been explained in greater detail in the introductory section.

Besides the confusion presented in the official reports and literature regarding the terminology what is the benefit of using the concept of soft skills in general, and specifically in the area of professional consulting for employment support?

In response to the improvement proposal, an explanation of the benefit of using the concept of soft skills in general and specifically in professional consulting for employment has been introduced in the text (introduction).

It is not clear why the professional “counselors are mostly people with limited economic resources and at risk of social exclusion”. Is this a specific circumstance of your sample?

Professional counselors are not at risk of social exclusion, but they work with people who show a risk of social exclusion due to being long-term unemployed, having economic and sociocultural difficulties and/or integrating into the business system.

Following the advice of the editor, we have provided a broader explanation at the end of section 2.1.

To help the reader to fully understand your sample characteristics you may add a table that clarifies the characteristics of the sample that provide complete data and the part that drops out. It would be informative to add also the observed frequency.

As suggested by the reviewer to clarify the characteristics of the sample, we have added a table at the end of section 2.1 that reflects the most significant and remarkable data.

In your interview protocol you had already chosen 24 soft skills it will be important to explore this option a bit further in the introduction.

Following the advice of the reviewer, an explanatory paragraph has been added to the introduction to clarify the selection of soft skills and to further explain the selection and classification.

Were the participants asked at any moment of the interview about skills or other aspects that they thought critical for employment?

To answer this question, an explanatory paragraph on the structure of the interviews has been added in section 2.2.

When the table is designed by the authors it is not probably necessary to add the following information in the legend: “Source: by authors”.

Thank you for the suggestion. We have eliminated this legend.

In the results when you write “There was an increase in these skills over the five-years study period” you meant in the perception of the importance of these skills?

This section refers to the perceptions of the counselors and the users. We have clarified this part.

It will help to read the quotations if you distinguish between the text and the participant to whom the statement belongs. It may also help to separate the quotation by skill, like putting all the quotations of communication, and then decision making, and so on.

As the reviewer suggested, citations have been included throughout the text to endorse the explanation; in addition, we have separated the citations by ability, and we have ordered all the communication citations as well as the decision-making citations.

Table 1 is displaying results, so it maybe be better to be put in the results section.

As suggested by the reviewer, the previous Table 1 (currently Table 2) has been included in the results section.

It is not clear how the mean values presented in the table were collected. Were the participants asked to rate the skills?

In each of the interviews conducted, the interviewee was asked to assess the level of soft skills following two criteria: 1) the importance of soft skills in the orientation process and 2) how this competence influences the level of employability of users.

We have provided this clarification in section 2.2.

It would be useful to add the results obtained when dividing the consolers by gender and also the rationale for trying to acknowledge these differences.

Following the advice, we have included the results (section 3) of the interviews conducted to endorse the coherence between what the counselors said and what we found in the literature.

The result section would be improved by separating the literature review and some explanation, limitations and reflections that can be placed in the introduction or the discussion and conclusion sections.

As suggested, in the results section, the literature review has been separated, and some explanations, limitations and reflections have been placed in the discussion and conclusion section.

In the results (3.1) you state that based on what the consulters shared during the different interviews it is proposed a process for promoting soft skills. Is it right? When you write “For these situations, they propose the following steps”, may you give the reader support from interviewer’s quotations?

Following the advice of the reviewer, comments from the counselors who supported the statements and good practices have been added.

The whole sub-section “Good Practices among Counselors for Developing Decision-Making and Communication in Counseling Processes” need to be supported in evidence from the data collection process otherwise it seems that is based in the literature review that you are citing thorough the text.

As suggested by the reviewer, we have incorporated evidence that supports the good practices section.

It is important to discuss, in the last section of your paper, the implication of your results and also the limitations and the future venues of research you are opening with this results.

The implications of the results, the limitations of this study and future research prospects have been included in the last section.
